# In Vivo Validation of Novel Synthetic *tbp1* Peptide-Based Vaccine Candidates against *Haemophilus influenzae* Strains in BALB/c Mice

**DOI:** 10.3390/vaccines11111651

**Published:** 2023-10-27

**Authors:** Naseeha Bibi, Amtul Wadood Wajeeha, Mamuna Mukhtar, Muhammad Tahir, Najam us Sahar Sadaf Zaidi

**Affiliations:** 1Vaccinology and Therapeutics Research Group, Department of Industrial Biotechnology, Atta Ur Rahman School of Applied Biosciences (ASAB), National University of Sciences and Technology (NUST), Islamabad 44000, Pakistan; naseehaqureshi@gmail.com (N.B.); awwajiha81@gmail.com (A.W.W.); mamunamukhtar@yahoo.com (M.M.); 2Department of Plant Biotechnology, Atta Ur Rahman School of Applied Biosciences (ASAB), National University of Sciences and Technology (NUST), Islamabad 44000, Pakistan; gullsbs@gmail.com

**Keywords:** *H. influenzae*, *tbp1* peptides, adjuvants, indirect ELISA, BALB/c mice, IgG antibodies, peptide antigens

## Abstract

*Haemophilus influenzae* is a Gram-negative bacterium characterized as a small, nonmotile, facultative anaerobic coccobacillus. It is a common cause of a variety of invasive and non-invasive infections. Among six serotypes (a–f), *H. influenzae* type b (Hib) is the most familiar and predominant mostly in children and immunocompromised individuals. Following Hib vaccination, infections due to other serotypes have increased in number, and currently, there is no suitable effective vaccine to induce cross-strain protective antibody responses. The current study was aimed to validate the capability of two 20-mer highly conserved synthetic *tbp1* (transferrin-binding protein 1) peptide-based vaccine candidates (*tbp1*-E_1_ and *tbp1*-E_2_) predicted using in silico approaches to induce immune responses against *H. influenzae* strains. Cytokine induction ability, immune simulations, and molecular dynamics (MD) simulations were performed to confirm the candidacy of epitopic docked complexes. Synthetic peptide vaccine formulations in combination with two different adjuvants, BGs (Bacterial Ghosts) and CFA/IFA (complete/incomplete Freund’s adjuvant), were used in BALB/c mouse groups in three booster shots at two-week intervals. An indirect ELISA was performed to determine endpoint antibody titers using the Student’s t-distribution method. The results revealed that the synergistic use of both peptides in combination with BG adjuvants produced better results. Significant differences in absorbance values were observed in comparison to the rest of the peptide–adjuvant combinations. The findings of this study indicate that these *tbp1* peptide-based vaccine candidates may present a preliminary set of peptides for the development of an effective cross-strain vaccine against *H. influenzae* in the future due to their highly conserved nature.

## 1. Introduction

*Haemophilus influenzae* is a Gram-negative bacterium commonly found in oral and nasal cavities inhabiting the upper respiratory tract of 90% of adults [1]. The bacterium is classified into six serotypes (a–f) based on their capsule type. Usually, children under 5 years of age are more vulnerable to infections, and various studies revealed that systemic infections due to invasive *H. influenzae* are worldwide more common in young children [2,3,4,5]. Invasive infections include pneumonia, septicemia, otitis media, osteomyelitis, meningitis, septic arthritis, and epiglottitis [6,7,8,9,10,11]. In some of the studies, it has been shown that infections due to *H. influenzae* are secondary infections caused by viral influenza [12]. *H. influenzae* strains are generally part of the normal anaerobic flora in the respiratory tract but can also become facultative anaerobic and act as opportunistic pathogens in immunocompromised patients or those suffering from other viral infections [1]. Vaccination against *H. influenzae* infections is highly recommended, especially for immunodeficient individuals who are more vulnerable to infections [12]. Before the introduction of the Hib vaccine in the 1980s, type “b” Hib was the primary cause of bacterial meningitis [12,13,14]. Hib not only accounted for the majority of cases of bacterial meningitis, but it was also responsible for over 80% of invasive infections [15]. Currently, the FDA (Food and Drug Administration, Silver Spring, MD, USA) has only three approved vaccines (Pedvax HIB, ActHIB, and Hiberix, licensed for clinical use in 1989, 1993, and 2009, respectively) up to date against *H. influenzae* infections. However, these polysaccharide- or toxoid-based conjugate vaccines do not confer complete protection against all *H. influenzae* serotypes [16]. A variety of Hib-conjugated vaccines widely used against Hib strains failed to provide enough protection to children (younger than 18 months), who are the group most vulnerable to *H. influenzae* infections [1]. Despite the globally available Hib vaccine, the emergence of non-Hib and non-typeable Hib (NTHi) invasive infections has been reported from various regions across the world [1,4]. This is possibly due to the capsule-switching phenomenon [17] in the less virulent types making them more virulent [8,18], thus strongly suggesting the need for the development of novel, effective prevention strategies [4,18]. However, the availability and production cost of Hib conjugate vaccines is a challenge in low-income developing countries [19]. Reverse vaccinology approaches provide a way to overcome these infrastructure challenges and thus help in validating and evaluating the efficacy of putative vaccine candidates [20]. Immunoinformatics can analyze pathogen genomes to identify potential antigens, overcoming issues with culturing and in vitro antigenic expression [21]. Synthetic vaccines containing target epitopes can combat tumors and infectious diseases [22]. Based on modern bioinformatics approaches, various in silico antigenic vaccine candidates have already been reported to produce excellent and promising preclinical immune responses [21]. While designing epitope-based vaccines, epitope mapping, predictions of B- and T-cell epitopes, and the binding efficiency of these epitopes with MHC complexes are crucial steps, since these epitopic sequences bind with specific MHC motifs [22].

Peptide-based vaccines have gained immense interest in modern immunotherapy due to ease of production, low cost of reproducibility, safety without adverse effects, and minimum dosage [23,24]. These vaccines are easily accessible to the target antibodies [25] and ensure more effective cell-specific antigen immune responses [22,26]. In addition to these peptides, the use of suitable and potent adjuvants is highly recommended, as it can play a crucial role in enhancing the immunogenicity of weak antigens by increasing the duration and speed of the immune reaction [27] while at the same time decreasing the dose requirements and thereby lowering the overall cost of expensive immunizations [23,26,28]. 

Tbp1 (transferrin-binding protein 1) epitopes as peptide vaccine candidates against Hib infections were used in the current study. This protein from different bacteria (*H. influenzae*, *Actino-bacillus*, *pleuropneumonia*, and *N. meningitidis*) has already been cloned and sequenced and is believed to be antigenically related in most bacterial species [1,29]. It is also highly conserved, since the functional lack of these tbp domains drastically affects bacterial growth [30]. Correspondingly, the presence of anti-tbp1 and -tbp2 antibodies has been demonstrated in various studies [31]. These proteins not only play a vital role in the in vivo acquisition of iron from the host but are also involved in virulence and are attractive vaccine targets [32] and excellent cross-protective antigens against different bacteria [33,34]. 

In the current study, data for the selection and design of synthetic peptides were retrieved from the reference source [35], and peptide selection was based on the best scores for the various essential parameters predicted using immunoinformatics approaches. Before carrying out experiments on the peptides in a wet lab, molecular simulation studies were employed for the confirmation of the candidacy of various peptides as suitable vaccine candidates; then, two 20-Mer *tbp1* epitopes were designed and synthesized to be used as peptide vaccine candidates and were validated with in vivo analysis. The focus of this study was to determine specific humoral responses against the selected *tbp1* epitopes/peptides that were administered to BALB/c mice in three escalating doses extended over 50 days. The peptides were derived from regions of the *tb1* protein that are found to be highly conserved across different strains of *H. influenzae*. The time course of immune responses was evaluated over one- and half-month timelines/periods, and titers were determined by performing indirect ELISA protocols.

## 2. Materials and Methods

### 2.1. Scheme of Study

A general overview of the methodology followed is presented in Figure 1.

### 2.2. Prediction and Selection of Overlapping B-Cell and T-Cell Epitopes as Peptide Vaccine Candidates

In this study, the data for the selection of peptide epitopes for the experiments were taken from a previous study by the same research group [35], where the *H. influenzae* type “a” (NML-Hia, GenBank accession: NZ_CP017811) reference strain was selected for detailed immunoinformatic analysis by using advanced computational tools. The epitopic sequences were mapped corresponding to the linear sequences reported to be in the surface-exposed regions of the *tbp1* protein. The Elli Pro, BCPreds, and MHCpred servers were used for the prediction of B- and T-cell overlapping epitopic sequences, followed by predicting the binding affinity of these sequences with HLA DRB1*0101 alleles by performing molecular docking. The best epitopes for vaccine development were those that were conserved and broadly distributed among clinical isolates, with higher levels of immunoreactivity and surface exposition in the native antigen [28]. In our previous study, it was found that the selected epitopes showed all the mentioned characteristics [35].

### 2.3. Prediction of IFN-γ, IL4, and IL10 Cytokine-Inducing Ability of Peptides

Predicted selective overlapping B- and T-cell epitopes were further investigated for their cytokine-inducing capabilities, such as IFN-γ (Interferon-gamma), IL4 (Interleukin 4), and IL10 (Interleukin 10). These cytokines are believed to aid in the activation of cytotoxic T cells and macrophages. The IFN-γ induction capability of the predicted epitopes was determined by using an online server, IFNepitope (http://crdd.osdd.net/raghava/ifnepitope/ accessed on 23 August 2022), by choosing default parameters of motif and SVM (support vector machine) hybrid model algorithms with an SVM threshold value of 0.2 [36,37,38]. This model is considered a highly accurate approach to IFN-γ-inducing epitope predictions. Likewise, the IL4pred and IL10pred servers were used for the IL-4- and IL-10-inducing properties of overlapping B- and T-cell epitopes, respectively. These IL4pred and IL10pred predictions were based on the SVM method, and the default threshold values were set to 0.2, and −0.3, respectively. 

### 2.4. Molecular Dynamics (MD) Simulations of the Receptor–Ligand Complex

Molecular dynamics simulations allow for the exploration of conformational energy sites accessible to protein molecules, thereby providing links between protein structure and dynamics [39]. Dynamics simulations were conducted for the best-scoring docked complexes (based on binding energy scores of epitopes and DRB1*0101 allele) to determine the stability of epitope–receptor docked complexes. The MD simulation was performed using the iMODS web server, an easily accessible and fast server, for the determination of protein flexibility by considering eigenvalues associated with the complex. The iMODS server evaluates protein structure stability using normal mode analysis (NMA) to compute internal coordinates. It represents the stability in terms of a main-chain deformability plot, covariance matrix, B-factor values, and elastic network model [40]. 

### 2.5. Immune Simulation of Selected Peptide Antigens

The immune simulation study was conducted for the selected peptides to predict the immune response profile. Immune simulations were performed using the C-ImmSim v10.1 online server, which predicts real-life-like immune interactions using the PSSM (position-specific scoring matrix) method [38]. Predictions of epitopes and immune interactions are made on a machine learning basis on this server [41]. Simulations of three different anatomical compartments, including bone, the lymphatic organ, and the thymus, take place on the server, whereby one can assesses the production of antibodies, cytokines, and interferon after injection of the antigen [42]. Moreover, this also allows one to speculate on Th1 and Th2 responses. The selected epitopic regions were analyzed for their immune response at 1050 steps. Three injections of the antigenic peptides were introduced at 1, 21, and 42 simulation step values, where 1 step stands for 8 h. All other simulation parameters were left at default. 

### 2.6. Molecular Docking: Epitopes and H2-Ad MHC-II Mouse Allele Receptor Binding

To explore the interaction between the selected epitopes (*tbp1*-E_1_ and *tbp1*-E_2_) and the MHC class II allele receptor H2-Ad of mice (Mus musculus), molecular docking was performed. The PDB structure and IDs for the H2 allele receptor were obtained from the RCSB PDB database (https://www.rcsb.org/ accessed on 27 September 2023). The PEP-FOLD3 (https://bioserv.rpbs.univ-paris-diderot.fr/services/PEP-FOLD3/ accessed on 27 September 2023) online server was used to predict the linear structures of amino acid sequences for the homology modeling of the predicted epitopes. The ClusPro 2.0 online tool for protein–protein docking (https://cluspro.bu.edu/login.php?redir=/models.php accessed on 27 September 2023) was used to verify the anchoring. The H2 mouse allele was set as the receptor macromolecule, and peptide epitopes were set as ligands. The process was performed by submitting PDB model files of receptors and ligands. The docked complex was visualized in 2D and 3D using the Discovery Studio visualizer tool.

### 2.7. Peptide Design and Synthesis

Based on in silico analysis predictions, two tbp1 peptides were selected and used to evaluate the humoral responses elicited by these antigenic regions with in vivo analysis. The selected epitopes were analyzed and screened out based on the best characteristics and parameters, thus making them good targets [35]. The 20-mer *tbp1*-E_1_ synthetic peptide and *tbp1*-E_2_ were manufactured commercially (Molecular Biology Products, Karachi, Pakistan) with a purity level of >90%. Suitable solvents were selected for the dissolution of the lyophilized peptides after analyzing the peptides for their acidic and basic nature; hence, *tbp1*-E_1_ (acidic in nature) was dissolved in 0.1 M NH_5_CO_3_ (ammonium carbonate) and diluted with PBS to the final concentration of 1 mg/mL. Similarly, *tbp1*-E_2_ (basic in nature) was dissolved in water (filter-sterilized) up to the final concentration of 1 mg/mL and stored at −20 °C until further use. Instructions for analyzing and dissolving peptides from various online sources were followed. This included using the gene script peptide handling guidelines (https://www.genscript.com/peptide_solubility_and_stablity.html accessed on 12 January 2021), Expasy ProtParam tools (https://web.expasy.org/protparam/ accessed on 12 January 2021), and the guidelines provided by LifeTein for using synthetic peptides (https://www.lifetein.com/handling_and_storage_of_synthetic_peptides.html accessed on 12 January 2021).

### 2.8. Ethics Statement 

BALB/c female mice (4–6 weeks old) weighing 20–24 g reared in a pathogen-free environment were obtained from the ASAB animal house laboratory. Ethical approval was obtained from the Institutional Review Board (IRB) of Atta-Ur-Rahman School of Applied Biosciences (ASAB), NUST, Islamabad, Pakistan. Animal care and handling were performed according to standard protocols in a controlled environment at 24 °C temperature with a proper daily light cycle (12 h light/dark cycle). Mice were acclimatized for 10 days before the start of the experiment to minimize the suffering of the animals used and were regularly provided with fresh clean drinking water and feed ad libitum. Average food intake, water consumption, and behavior were observed regularly before the start of the experiment. 

### 2.9. Mice Immunizations and Grouping 

Animals (female BALB/c mice) were divided into 9 groups (8 mice/A, B, and C groups, 3 mice/control groups) (Table 1). Different combinations of 20-mer peptides (*tbp1*-E_1_ and *tbp1*-E_2_) with two different adjuvants, BGs (Bacterial Ghosts) and CFA/IFA (complete/incomplete Freund’s adjuvants), were used in each group. The BG adjuvants used in this study were prepared freshly for the experiments by using DH5α cells of *E. coli*.

CFA (0.5 mg/mL) was used in primary shots in groups A1, B1, and C1, followed by 1st and 2nd booster shots with IFA (1 mg/mL) in combination with *tbp1*-E_1_, *tbp1*-E_2_, and *tbp1*-E_1_+*tbp1*-E_2_, respectively, whereas in groups A2, B2, and C2, the BG adjuvant was used (1 mg/30 µL in PBS) in combination with *tbp1*-E_1_, *tbp1*-E_2_, and *tbp1*-E_1_+*tbp1*-E_2_ peptides, respectively. The peptide: adjuvant formulations were administered on days 1, 14, and 28, maintaining two-week intervals. Three groups, D1, D2, and D3 (*n* = 3), serving as controls, received only CFA/IFA, BG, and PBS subcutaneous injections, respectively, at the same intervals. In formulations of peptides and adjuvants, a 1:1 ratio was maintained for all the doses (Table 1). 

### 2.10. Pre-Immunization Procedure 

Small-size needles of 1 microliter were used for the subcutaneous (SC) injections of the doses. The minimum and maximum effective doses administered in each group were first optimized for safety purposes. Injection formulation was prepared before injecting by vortexing equal amounts of adjuvant and peptide solution in a 1:1 ratio. The animals were immunized via the subcutaneous (SC) route in the neck region with 120–150 µL of the formulations and observed twice a day after they were immunized for detecting changes in their behavior, signs of disease, or any adverse effects, and a weekly physical examination was made for monitoring the animals’ overall state of health. Their weight, body temperature, and food consumption were also monitored before beginning the immunization protocols and after immunizations every other day. 

Blood was drawn 2 weeks after the last immunization via cardiac puncture according to standard protocols (by sacrificing the animals after chloroform overdose). The blood was centrifuged at 6000 rpm for 10 min, and the supernatant was gently transferred into new tubes and stored at −20 °C. The serum separated from each blood sample was linearly diluted (2 times) with PBS from 500× to 64,000× for the determination of IgG titers using the indirect ELISA optimized protocol. The pooled sera from each group were used to evaluate differences in antibody titers in response to different combinations of peptides and adjuvants.

### 2.11. Detection of Humoral Response with ELISA

Antibody titers of all experimental and control groups were measured with an indirect ELISA employing *tbp1*-E_1_ and *tbp1*-E_2_ peptides as coating antigens. 

Antigens *tbp1*-E_1_ and *tbp1*-E_2_ were diluted to a final concentration of 5 µg/mL using 100 mM bicarbonate/carbonate coating buffer (pH 9.6). Wells of a PVC microtiter plate (SPL Life Sciences Co., Ltd. (Silver Spring, MD, USA)) were coated with 100 µL/well diluted antigen and incubated overnight at 4 °C. Coated wells were washed 3× with 200 µL of washing buffer, PBS-T (0.2% tween 20 in phosphate-buffered saline), followed by blocking with 200 µL of 5% *w*/*v* non-fat dry milk (skim milk) in PBS to block residual protein binding sites, and incubated for 1–2 h at 37 °C. The wells were washed again 3× with 200 µL of PBS-T washing buffer followed by adding 100 µL/well diluted serum with two-fold dilutions (500×, 1000×, 2000×, 4000×, 8000×, 16,000×, 32,000×, and 64,000×). The plate was covered and incubated at 37 °C for 1 h and washed again with 200 µL of PBS-T washing buffer. Then, anti-mouse HRP-conjugated antibody (Peroxidase Conjugated Affinity Purified anti-Mouse IgG (H & L) abm^®^ (Applied Biological Materials Inc.) Richmond, BC, Canada was diluted 1:20,000 times, and 100 µL of it was loaded into each well. The plate was covered and incubated at 37 °C for 2 h followed by washing 3X with PBS-T to remove unbound antibodies. A total of 100 µL of HRP substrate (TMB, Tetramethylbenzidines; High Sensitivity; abcam Waltham, USA) was loaded to achieve the reaction in the dark at 37 °C for 15 min. The reaction was stopped by adding 50 µL of 2 M H_2_SO_4_. The absorbance was measured at 450 nm using a microplate reader (Bio-Rad Magellan PR4100, A Bio-Red Laboratories Inc., Hercules, CA, USA) within 30 min of stopping the reaction.

### 2.12. Statistical Analysis

For the determination of endpoint antibody titers, the cutoff values were calculated from OD450 nm values of the negative control samples (group D3 with PBS subcutaneous injections, *n* = 3) for a 95% confidence interval by using the formula “Cutoff = $\bar{X}$ + SD*f* ” as described in [43], which utilizes the Student’s t-distribution method for the prediction of the upper cutoff limits. Background corrections were made by subtracting the means of blanks from the rest of the values. GraphPad Prism version 9.8.1 (GraphPad Software, San Diego, CA, USA) software tools were used for plotting the titration curves. Significant differences between the OD450 nm absorbance values of experimental and negative control groups were compared using the Mann–Whitney U test. A *p*-value of less than 0.05 was considered statistically significant. 

## 3. Results

### 3.1. Prediction and Selection of Overlapping B-Cell and T-Cell Epitopes as Putative Candidates for Peptide Vaccine

The current study aimed to validate in silico predictions with wet lab experiments. Briefly, from our source data [35], the proteome of the typeable strains of *H. influenzae* type “a” bacterium (NML-Hia; GenBank accession: CP017811) was used for detailed analysis. Proteins that were located in outer-membrane regions and were essential, virulent, and non-homologous were screened out for physicochemical analysis (molecular mass, pI value, functional annotation, estimated half-life, instability index, aliphatic index, GRAVY, and antigenicity scores for further shortlisting as potential vaccine candidates). Five proteins (peptidoglycan-associated lipoprotein (Pal OMP P6), glutamate dehydrogenase (GDH), transferrin-binding protein (Tbp1), type IV pilus biogenesis/stability protein (PilW), and porin OmpA) were shortlisted based on the above-mentioned parameters (Appendix A) and were further subjected for B- and T-cell overlapping epitope mapping using BCpred and MHCpred tools. Of these, the nine best-scoring overlapping B- and T-cell epitopes were selected based on BCpred scores > 0.8 and their ability to bind with the maximum number of MHC alleles. Molecular docking was performed by using Pyrex Docking tools between the DRB1*0101 allele (receptor) and the selected proposed nine peptides/epitopes (ligands). The HLA DRB1*0101 allele was chosen for docking, as it contributes to enhanced immunogenic activation based on the efficient antigen-presenting capability of the proteins and is the most frequently bound component. The Tbp1 epitopes with the highest binding energy scores (−7.8 kcal/mol, −7.2 kcal/mol, and −6.8 kcal/mol) had the best binding score after docking [35] (Appendix A). Based on predictions, the two best-scoring epitopes (*tbp1* epitopes) of the total nine were chosen for commercial synthesis and subsequently for use as peptide vaccine candidates in the BALB/c mouse model for the detection of immune responses.

### 3.2. Prediction of Cytokine-Inducing Epitopes 

Following the prediction of scores for various parameters, the cytokine-producing abilities of the chosen B- and T-cell overlapping epitopes were thoroughly examined. The IFNepitope server ranks the inserted antigenic sequence in positive or negative order by using the support vector machine (SVM) and motif hybrid method based on their potential to induce IFN-γ. Positive scores for IFN-γ induction were predicted by selecting IFN-γ versus non-IFN-γ. Similarly positive results were obtained for IL4- and IL10-inducing features, and epitopes were ranked as inducers of the mentioned interleukins (Table 2). Based on prediction results, the selected epitopes were ranked for their efficacy as cytokine inducers, which play a critical role in the activation of immune cells.

### 3.3. Molecular Dynamics Simulation

The molecular interaction in the tbp1 vaccine peptide (ligand)-and-HLA DRB1*0101 (receptor) docked complex was screened for protein stability, deformity, and B-factor mobility by using the iMODs server (Figure 2).

The MD simulation of the interacting docked complex was based on the geometry coordinates between ligand–receptor superimposed protein complexes. Deformation of the protein complex was easier at a lower eigenvalue, indicating easier deformation of the complex, and its values were directly proportional to the energy needed for structure deformation (Figure 2a). The eigenvalue was found to be 3.046751 × 10^−6^, indirectly indicating that the complex could successfully activate the immune mechanisms to destroy the foreign antigens. The variance map, which is inversely proportional to the eigenvalue, is displayed in Figure 2b. The purple-colored bars in the variance graph represent the individual variance, and cumulative variance is depicted by green-colored bars. In Figure 2c, the B-factor values calculated using NMA (normal mode analysis) are proportional to the RMS (root mean square), which measures the level of uncertainty for each individual atom. In Figure 2d, the main-chain deformability is shown, and the locations with hinges are regions with high deformability. Figure 2e indicates correlations between the pairs of residues as a covariance matrix (red: correlated; white: uncorrelated; blue: anti-correlated). The elastic network model suggesting the connection between atoms and darker gray regions of springs represents the stiffer regions, as shown in Figure 2f.

### 3.4. Immune Simulation Studies

The immune simulation of the best-selected tbp1 epitopes was performed using the C-ImmSimm server, which predicts immune interactions and the generation of adaptive immunity. The findings of the in silico immune simulations of the peptides are provided in Figure 3. Immune simulation outcomes confirmed consistency with real immune reactions for a duration of 60–90 days in terms of an increase in antibody titer after booster shots and a decrease in antigen levels, thus indicating its clearance (Figure 3a–i). This study focused on the ability of the peptides to stimulate immune cells like dendritic cells (DCs), natural killer cells (NKs), B cells, helper T cells (HTCs), cytotoxic T cells (CTCs), and immunoglobulins (IgGs). The simulation study revealed a sharp increase in IgG concentrations levels, indicating a primary immune response against the selected antigenic fragments, as a steady increase in the production of more antibodies (IgG, IgG1, IgG2, IgG + IgM) was detected after each dose of the antigen (Figure 3a), while rapid clearance of the antigen was observed. The results of the production of interleukins and cytokines indicate a strong response of IFN-γ and IL-2 to the target vaccine (Figure 3b). Also, the antigen was seen to initiate the production of increased and maintained B- and T-cell populations (Figure 3c). A significant increase in the number of B-cell memory and active T-cell populations can be seen in Figure 3d. These observations suggest that there was a rise in robust secondary immune responses targeting the epitopes. Figure 3e–h show increased DC and NK-cell populations, indicating APCs as good antigen-presenting cells for immunogenicity response. To sum up, the study on immune simulation showed that the proposed peptide-based vaccine candidates have the potential to provide effective cross-strain immunogenic protection against various strains of *H. Influenzae*. This is because they are anticipated to stimulate the production of high levels of IgGs (immunoglobulins), active B- and T-cell populations, and cytokines. Thus, these results display that the peptide antigens could induce the immune response very well and provide the basis for immunity against influenza-associated infections. 

### 3.5. Docking of H2-Ad Mouse Allele and Selected Epitopes 

ClusPro 2.0 was used for molecular docking to determine peptide binding groove affinity between H2 allele and peptides. Figure 4 shows the active pocket residues from the beta chain of the H2 mouse allele for *tbp1*-E_1_ and *tbp1*-E_2_ peptide ligands. The Discovery Visualization offline tool was used to visualize binding patterns and active-site residues, which indicated excellent binding ability of the peptide with the beta chain of the selected allele (Figure 4a–d).

### 3.6. Detection of Humoral Response with ELISA

A semi-quantitative indirect ELISA was used as it confirms positive or negative results, in addition to the comparison of IgG target antibody levels against the coated antigens, since the level of absorbance directly corresponds to the level of target protein concentration (Figure 5).

Sera of the immunized BALB/c mice receiving different doses of antigen: adjuvant combinations were tested for specific antibody IgG titers. The highest serum dilutions at which the OD450 nm mean absorbance was equal or at least two times more than the mean absorbance value of the negative control were considered endpoint titers. Endpoint antibody titers reached 1:16,000, and the results are expressed as the titer being equal to the reciprocal of the serum dilution (Figure 5a). Statistical analyses were performed with the Mann–Whitney U test, and *p*-value <0.05 was considered significantly different in all cases. Subcutaneously inoculated groups (A1–C2) showed significantly higher titers (*p* < 0.05) compared with control groups (D1–D3). However, the groups inoculated with adjuvants alone showed insignificant antibody titers (*p* > 0.05) compared with negative control group D3, which was inoculated with PBS. Statistical differences between groups A1 and A2 (towards *tbp1*-E_1_ peptide), groups B1 and B2 (towards *tbp1*-E_2_), and C1 and C2 (*tbp1*-E_1_+E_2_) revealed insignificant differences in the ODs for different groups. However, the endpoint antibody titer for groups A1 and A2 was more precisely 1:8000, while for the remaining groups, it reached 1:16,000. Similarly, in groups A1 and A2, the absorbance values were lower at OD450 nm compared with the OD450 nm absorbance values for the remaining groups (Figure 5b). 

Mice in groups C1 and C2 that received *tbp1*-E_1_+E_2_ with adjuvants produced high absorbance values for OD450 nm and showed high endpoint antibody titers in comparison to those groups that received *tbp1*-E_1_ or *tbp1*-E_2_ alone with either of the adjuvants. Similarly, groups B1 and B2’s overall response both in terms of titer and absorbance values for the same dilution parameter was higher compared with groups A1 and A2 (Figure 5a,b). As shown in Figure 5b, the Mann–Whitney U test showed a statistical difference for group B1, with a *p*-value <0.05. The calculated *p***-value was 0.002, while for the other groups, the *p**-value was 0.02 compared with negative control group D3. Higher absorbance values for ODs450 nm were observed in groups where BGs were used as an adjuvant in comparison to CFA/IFA. Similar results with low levels of antibody detection were observed for control groups D1 and D2, where CFA/IFA and BGs were used alone as control, respectively; the OD value for group D2 was slightly higher than that for D1. The antibody levels in the control groups (D1 and D2) that received CFA and BGs, respectively, were much lower compared with the endpoint titers of the experimental groups (A1–C2). The statistical differences between them were insignificant, as shown in Figure 5b, where *p* = 0.1. In summary, the response in groups with BGs (C2, B2) was significantly higher than that in the CFA group. Additionally, the combined use of both epitopes resulted in a better response. 

## 4. Discussion

This study aimed to validate two putative peptide vaccines with wet lab experiments. The vaccines were predicted based on in silico analysis techniques and were designed to determine immune responses against highly conserved epitopic regions of *H. influenzae*. This bacterium is known as a human-restricted bacterium, responsible for causing invasive infections in children and immunocompromised adults. Infections due to serotype “b” have been highly prevented after Hib vaccinations; however, the emergence of infections by other serotypes and non-typeable isolates (NTHi) has been observed, as Hib vaccination was found to be ineffective in deliberating protection against other serotypes [44], thereby prompting the need for developing and establishing novel prophylactic strategies against *H. influenzae* [4]. This study, therefore, mainly focused on using a peptide-based vaccine design against *H. influenzae* using data from the previous study by the same research group [35]. The key objective was to design and validate a peptide-based vaccine composed of highly conserved epitopic sequences to induce cross-strain effective immune responses; thus, we aimed at the validation and translation of the *In-silico* findings into in vitro and in vivo experimental setups. Peptide vaccine development strategies take advantage of advanced immunoinformatics tools, such as epitope mapping, for the prediction and identification of ideal vaccine candidates [45], their interactions with suitable receptors via docking, molecular dynamics (MD) simulations to check the stability of the predicted structures, and immune simulation studies that can be translated through their validation and confirmation into in vitro assays (e.g., ELISA for detection of humoral responses). Only two *tbp1* epitopes (consisting of B- and T-cell overlapping sequences) of the nine predicted epitopes [35] were used in this study based on the highest binding energy scores of −6.8 kcal/mol and −7.2 kcal/mol and the highest antigenicity scores of 1.3 and 0.9, respectively. Since the experiments were performed in mice, the H2-Ad mouse allele (MHC-II allele) was also considered for molecular docking analysis. ClusPro 2.0 was used to check the binding interactions between selected epitopic regions, and the surface binding sites on the beta chain of the H2 allele receptor. Various studies have reported *tbp1* as a potent vaccine candidate acting as an immunogen in several Gram-negative bacteria [33]. To further enhance the immune potential of epitopes, BGs and CFA/IFA were used in this study. The MD simulation was performed to comprehend peptides’ stability and flexibility, as it has the potential to help in a better understanding of protein structures and firmness. Additionally, high levels of IFN-γ and interleukin secretion, and a long-lasting cellular response were predicted with in silico immune response simulations, which emphasizes the possibility of inducing an effective immune response. In many different studies, the usefulness of bioinformatics approaches for the development of effective vaccine candidates against different pathogens has been demonstrated [28,41,42,46,47,48]. Positive results for the induction of the well-known broadly expressed cytokines IFN-γ, IL4, and IL10 were predicted for the peptides. This cytokine has a crucial role in preventing inflammation [38], immunosuppression, and the development of suitable effector T-cell responses [49]. For the determination of the immune potential of the synthetic *tbp1* peptides, the indirect ELISA technique was optimized. Various studies have demonstrated the importance of developing an indirect ELISA, which is not only advantageous in finding out the potential of immunogens (peptides and other vaccine antigens) but is also cost-effective [50,51]; therefore, the development of optimized indirect ELISA protocols for the detection and determination of antibodies in different infections is crucial [12]. All BALB/c mouse groups that received combinations of peptides/adjuvant vaccination produced high antibody titers in comparison to the control group. Statistical analyses were performed for *p* < 0.05 taken as statistically significant. The highest titer levels and high absorption values were observed for group C2, where both peptides (*tbp1*-E_1_+E_2_) were used together in combination with BGs as an adjuvant. BGs were freshly prepared from DH5α cells; they consist of only empty cell envelopes and are considered to have a role in the modulation of Th1/Th2 mixed response to more dominant Th2 response [52]. Due to these adjuvant properties, BGs act as an efficient carrier system for external antigenic sequences [53]. In group C1, where CFA/IFA was used in combination (*tbp1*-E_1_+E_2_), the absorption values were slightly lower. Likewise, in comparisons between groups A (A1 and A2) and B (B1 and B2), the latter showed significant differences in terms of high titers and high absorption values. The results revealed that the groups where BGs were used (A2, B2, and C2) showed high absorbance values for OD450 nm as compared with CFAs. Significant statistical differences (*p* < 0.05) were found for all groups (A1–C2) in comparison to negative control group D3. In several considerable studies, the effectiveness of BG adjuvants has demonstrated their potential for the production of proinflammatory cytokines and subsequent activation of potent immune responses [54]. These results are an excellent match in various parameters (e.g., VaxiJen score, binding energies, and other physicochemical values) determined for both peptides using in silico approaches, thus demonstrating the validity of immunoinformatic-based predictions for *tbp1* peptides. This study is the first to reveal the immune response-inducing abilities of two epitopic regions from the *H. influenzae* strain. These tbp1 antigens, which have been predicted with in silico approaches, are highly conserved across various strains of *H. influenzae*. Furthermore, they have not been considered vaccine candidates in previous reports. Moreover, various studies have reported the presence of natural antibodies against *tbp1* antigens in various cases, which further supports their use as future vaccine candidates against *H. influenzae* infections. According to this study, tbp1 peptides show promise as potential vaccine candidates for fighting against *H. influenzae* infections. This could result in the development of vaccines that can effectively combat various strains of the infection. However, additional research is necessary to validate these discoveries, including challenge studies to assess the level of protection provided by the vaccine in future investigations.

## 5. Conclusions

To summarize, this study proposes a cost- and time-efficient method to identify suitable vaccine antigens for potential use in wet lab experiments. The approach of validating in silico predictions in wet labs can also be extended to discovering new vaccine candidates for other types of pathogenic bacteria. These *tbp1* antigens can be an alternative to serotype-specific Hib-conjugated vaccines used against Hib infections, which fail to confer protection against other serotypes of *H. Influenzae*, considering these to be highly conserved over major typeable serotypes. The findings of this study suggest that tbp1 peptides could be used as vaccine candidates against *H. influenzae* infections, paving the way for the development of effective cross-strain vaccines. However, these results need more validation through challenge studies for the determination of protective efficacy in future studies.

## Figures and Tables

**Figure 1 vaccines-11-01651-f001:**
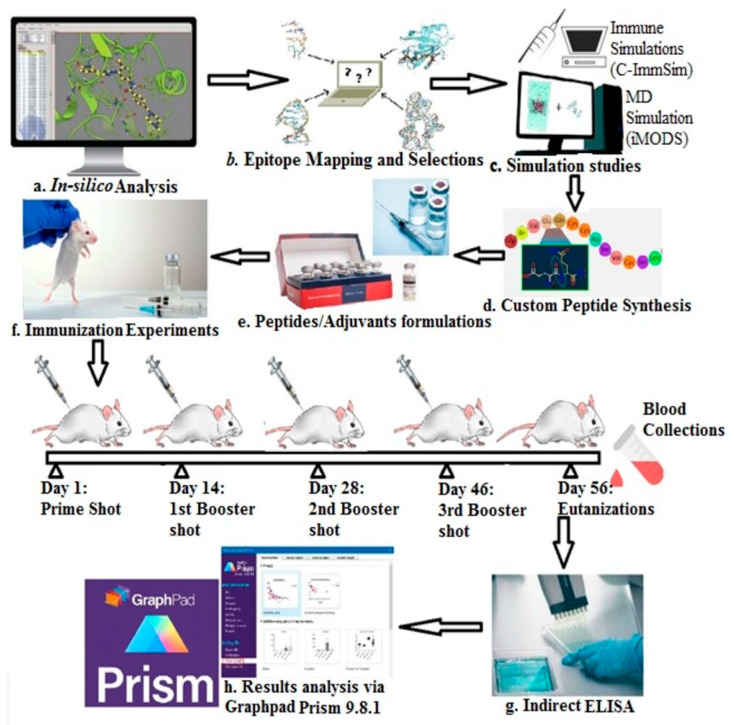
Abstract diagram showing the basic methodology followed.

**Figure 2 vaccines-11-01651-f002:**
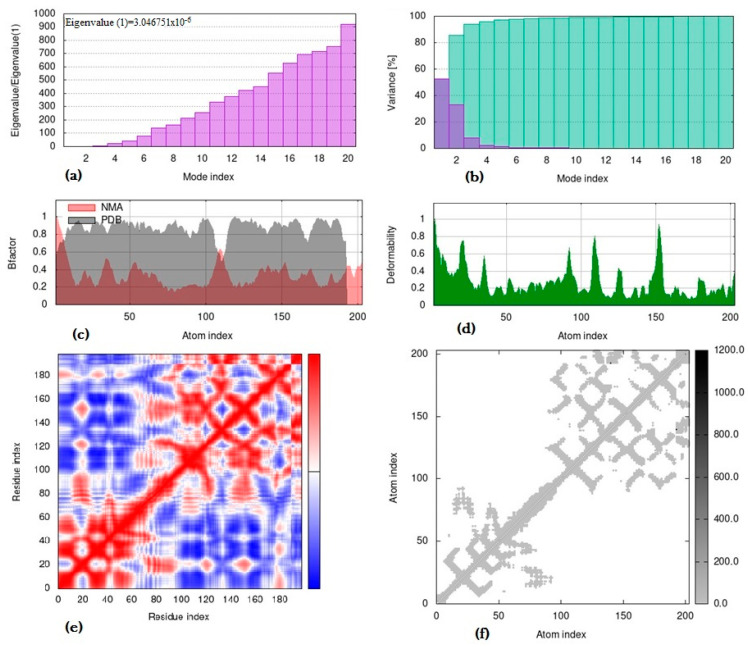
Molecular dynamics simulation of peptide–DRB01010 docked complex, showing (**a**) eigenvalue; (**b**) variance; (**c**) B-factor; (**d**) deformability; (**e**) covariance matrix, which indicates coupling between pairs of residues (red), and uncorrelated (white) or anti-correlated (blue) motions; and (**f**) elastic network analysis, which defines which pairs of atoms are connected by springs.

**Figure 3 vaccines-11-01651-f003:**
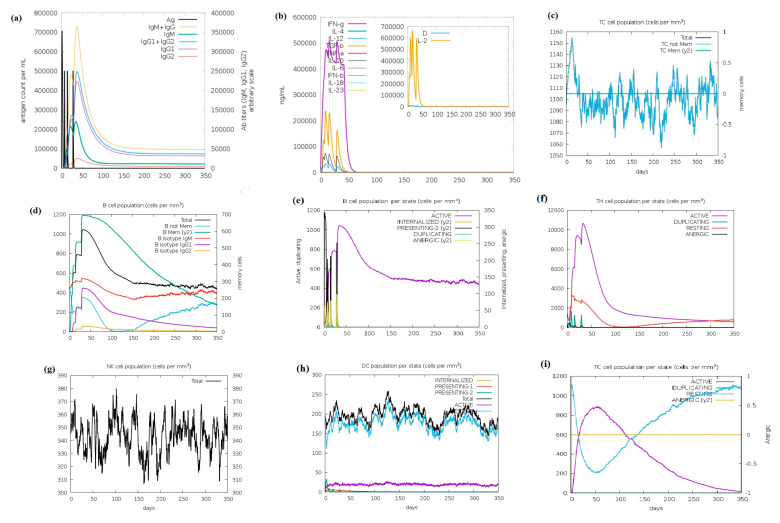
The immune simulation results were generated with the online tool C-Immism. The graphs show (**a**) immunological reactions in terms of IgG titers (primary, secondary, and tertiary); (**b**) induction of cytokines and interleukins, where the inset plot shows danger signal together with leukocyte growth factor IL-2; (**c**) cytotoxic T-cell population; (**d**,**e**) B-cell population; (**f**) helper-T-cell population; (**g**) natural killer cells; (**h**,**i**) dendritic-cell and T-cell populations per state. (Legends: Act = active; Intern = internalized the Ag; Pres II = presenting on MHC II; Dup = in the mitotic cycle; Anergic = anergic; Resting = not active.)

**Figure 4 vaccines-11-01651-f004:**
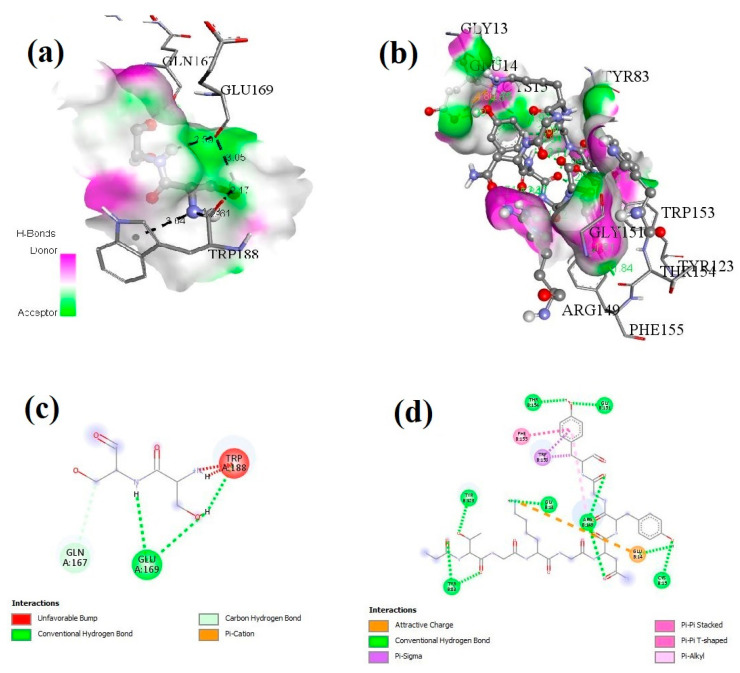
Spatial representation of the dock molecule (ligands and H2 allele receptor). (**a**,**b**) Active pocket residues on beta chain of receptor molecule, showing the *tbp1*-E_1_ (**a**) and *tbp1*-E_2_ (**b**) ligand binding sites. (**c**,**d**) Two-dimensional structure of docking ligand and receptor molecule.

**Figure 5 vaccines-11-01651-f005:**
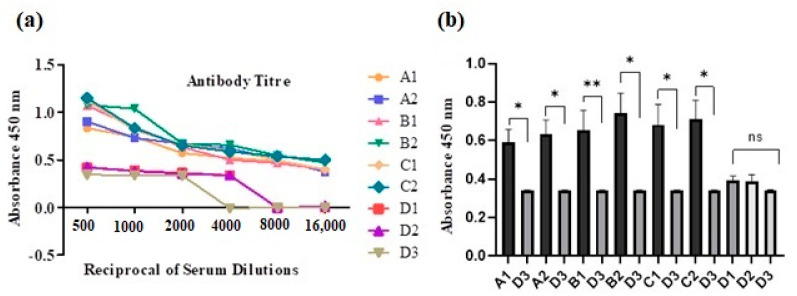
IgG antibody response in BALB/c mice against tbp1 peptides and adjuvants, inoculated via subcutaneous route. OD values of peptides vs. controls are depicted on the *Y*-axis. Absorbance was measured at 450 nm. Data collected from 6 mice per group are expressed as ± standard error of the mean (SEM) at a serum dilution of 1:16,000. (**a**) Endpoint IgG antibody titers in sera of mice primed subcutaneously with peptide + CFA/BG and boosted with peptide + IFA/BG are shown; the titer is represented as reciprocal of serum dilutions corresponding to 1:16,000. (**b**) Statistical significance was determined with the Mann–Whitney U test, and *p*-value < 0.05 was taken as significant. The asterisks indicate significant statistical difference. *p*-Values are indicated as follows: * *p* = 0.02, ** *p* = 0.002, and “ns” *p* = 0.1 when compared with negative control group D3.

**Table 1 vaccines-11-01651-t001:** Experimental design for mouse immunization and groupings.

Group	Subgroup	Day 1 Prime Shot	Day 14 1st Booster Shot	Day 28 2nd Booster Shot	Day 42
A	A_1_ (*n* = 8)	(*tbp1*-E_1_)+CFA	(*tbp1*-E_1_)+IFA	(*tbp1*-E_1_)+IFA	Euthanization and blood sampling
	A_2_ (*n* = 8)	(*tbp1*-E_1_)+BG	(*tbp1*-E_1_)+BG	(*tbp1*-E_1_)+BG	
B	B_1_ (*n* = 8)	(*tbp1*-E_2_)+CFA	(*tbp1*-E_2_)+IFA	(tbp1-E_2_)+IFA	
	B_2_ (*n* = 8)	(*tbp1*-E_2_)+BG	(*tbp1*-E_2_)+BG	(tbp1-E_2_)+BG	
C	C_1_ (*n* = 8)	(*tbp1*-E_1_+E_2_)+CFA	(*tbp1*-E_1_+E_2_)+IFA	(*tbp1*-E_1_+E_2_)+IFA	
	C_2_ (*n* = 8)	(*tbp1*-E_1_+E_2_)+BG	(*tbp1*-E_1_+E_2_)+BG	(*tbp1*-E_1_+E_2_)+BG	
D	D_1_ (*n* = 3)	CFA	IFA	IFA	
	D_2_ (*n* = 3)	BGs	BGs	BGs	
	D_3_ (*n* = 3)	PBS	PBS	PBS	

Tbp1 (transferrin-binding protein 1), E_1_ (Epitope 1), E_2_ (Epitope 2), *n* (number), CFA (complete Freund’s adjuvant), IFA (incomplete Freund’s adjuvant), BGs (Bacterial Ghosts), PBS (phosphate-buffered saline).

**Table 2 vaccines-11-01651-t002:** Cytokine-inducing epitopes.

S. No.	Peptide/Epitope	IFN-γ	IL4	IL10
1.	*tbp1*-E_1_	Positive	Inducer	Inducer
2.	*tbp1*-E_2_	Positive	Inducer	Inducer

IFN-γ: Interferon-gamma; IL4: Interleukin 4; IL10: Interleukin 10.

## Data Availability

All data that this study is based upon are available from the corresponding author upon request.

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
