# Peer review of "In Vivo Validation of Novel Synthetic tbp1 Peptide-Based Vaccine Candidates against Haemophilus influenzae Strains in BALB/c Mice"

_vaccines, 2023, doi:10.3390/vaccines11111651_

Round 1
Reviewer 1 Report (Previous Reviewer 2)
I asked that “Authors chose HLA-DRB1*0101 allele for selection of T-cell epitope and in silico docking experiment. In this manuscript, authors use BALB/c mice for immunization. Authors should examine whether the peptide antigens can bind to H2-Ad allele.” Authors’ response to this comment is not satisfactory. Only MHC class II molecule expressed in BALB/c mice is H2-Ad. It is easy to check the peptide used in this study can bind to H2-Ad. By using an epitope prediction algorithm, both Tbp1 E1 and E3 peptides could not bind HLA-DRB1*0101 and H2-Ad.
Author Response
Please see the attachment

Reviewer 2 Report (Previous Reviewer 1)
MS is in acceptable form.
Author Response
Please see the attachment

Reviewer 3 Report (New Reviewer)
Title: In-vivo Validation of Novel Synthetic tbp1 Peptide-Based Vac- 2 cine Candidates against Haemophilus influenzae Strains in 3 BALB/c Mice
ID No.: vaccines-2555674
Haemophilus influenzae causes diseases in both adult and young. However, children are more susceptible to infections. Though treatment is available but vaccination is one of the very effective tools to control the disease. This manuscript describes both dry and wet lab activities to validate peptide-based vaccine.
The manuscript is well written. Minor corrections are suggested below and also marked on the manuscript. The manuscript may be accepted for publication after suggested modifications.
Line 40: delete comma; add ‘and’ between the marked words
Line 45: influenza instead of influenzae
Line 46: delete the word organisms
Line 48-52: Recheck the sentences and relevant references
Line 67-70: Recheck the sentence.
Line 94: Check the highlighted words.
Line 189: Delete the highlighted words.
Line 192: Do italic highlighted words.
Line 196: In control group not 8 mice
Line 221: Check the highlighted word
In Figure 4a- OD values are mentioned as Antibody response.
4b- Titre is shown. Actually titre is calculated from OD values.
What is the importance of showing OD value (Fig 4a) is not clear to me rather arising confusion. Need further clarification.
I have found yellow highlighted text throughout the manuscript I have downloaded from the system. Those were not done by me.
Reviewer

Author Response
Please see the attachment

Reviewer 4 Report (New Reviewer)
A well-designed work. A well-written paper.
The study did not include challenge experiments. Anyway, based on the observed immune response, can the authors anticipate satisfactory protection against H. influenzae from their vaccine candidates?
Can the authors anticipate cross-strain protection from their candidates (based on the observed immune response)?
What is the difference between Fig 4a and 4c? Can 4a be removed?
The sentence in lines 394 though 399 is way too long and incomplete. Need to be corrected. Please closely check for similar editorial defects throughout the paper.
Quality of English is satisfactory.
The sentence in lines 394 though 399 is way too long and incomplete. Need to be corrected. Please closely check for similar editorial defects throughout the paper.
Author Response
Please see the attachment

Reviewer 5 Report (New Reviewer)
Bibi et al. reported on the in-silico design and screening, as well as the in-vivo evaluation of peptide-based vaccine candidates against Haemophilus influenzae. The manuscript demonstrates a clear and professional use of the English language, making it easy to follow. However, some of the Figures and Tables need improvement, as outlined below:
(1) In Figure 3, the authors should consider adjusting the spacing between subfigures to enhance the visibility of axis labels. Currently, it is challenging to distinguish the axis labels among the subfigures.
(2) Supplementary Table 2: The unit of IC50 data appears confusing and requires clarification.
(3) In the Discussion section, the authors may want to address the limitations and future directions of this study. For example, can a high antibody titer through ELISA conclusively indicate the promise of these peptides as vaccine candidates? Additionally, the authors may consider noting that this project only included in-vitro toxicity studies.
Author Response
Please see the attachment

This manuscript is a resubmission of an earlier submission. The following is a list of the peer review reports and author responses from that submission.
Round 1
Reviewer 1 Report
Comments:
1. Why only three cytokines viz. IFN-γ, IL4 and IL10 selected for studying the Cytokines Inducing Ability of peptides? Explain.
2. Explain the reason behind choosing 1, 21, and 42 simulation step value for the injections of antigenic peptides in Immune Simulation analysis.
3. The paper suggests that further validation through challenge studies is required to determine the protective efficacy of the tbp1 peptide-based vaccine candidates. Also, the study was conducted on female BALB/c mice, and the results may not be generalizable to other animal models or humans. Why the protective efficacy has not been studied in this paper?
4. How were the acidic and basic natures of tbp1-E1 and tbp1-E2 peptides respectively evaluated?
5. Justify the reason for choosing 9 different groups for immunizing mice with antigenic peptides and different adjuvants.
6. Explain the reason of choosing 0.5mg/ml, 1mg/ml, and 1mg/30μl in PBS as the doses of CFA, IFA, and BG respectively?
Not applicable
Reviewer 2 Report
The topic is interesting. But the manuscript has several critical issues.
Major comments
What do yellow-labelled parts mean? Is this manuscript a revised manuscript, not a primary manuscript?
1. Authors chose HLA DRB1*0101 allele for selection of T-cell epitope and in silico docking experiment. In this manuscript, authors use BALB/c mice for immunization. Authors should examine whether the peptide antigens can bind to H2-Ad allele.
2. Authors showed only antibody titers against two peptide antigens used. Authors should examine the antibody class and subclass detected. The proportion of IgG1 and IgG2a may help to evaluate the immune responses.
3. Authors should examine T-cell responses against peptide antigens with wet lab experiment.
4. In Materials and Methods section, authors do not clearly describe amounts of antigens used for immunization. The total amounts of peptides used should be described.
Minor comments
Reference numbers in text should be parenthesized.
Some references seem to be inappropriate. Check them again.
Supplementary Table 2. Why are Tbp1 E1 and E3 peptide bold? Were not Tbp1 E1 and E2 used for experiments?
English should be improved. Several typographical and careless mistakes were found.
The examples are shown below.
Line 27. Students’ should be “Student’s”.
Line 46. “anerobic” reads “anaerobic”.
Line 91. “effects” reads “affects”.